# Sodium Intake and Proteinuria/Albuminuria in the Population—Observational, Cross-Sectional Study

**DOI:** 10.3390/nu13041255

**Published:** 2021-04-11

**Authors:** Massimo Cirillo, Pierpaolo Cavallo, Enrico Zulli, Rachele Villa, Rosangela Veneziano, Simona Costanzo, Sara Magnacca, Augusto Di Castelnuovo, Licia Iacoviello

**Affiliations:** 1Department of Public Health, University of Naples Federico II, 80131 Naples, Italy; enrico.zulli@gmail.com; 2Department of Physics, University of Salerno, 84084 Fisciano, Italy; pcavallo@unisa.it; 3Istituto Sistemi Complessi, Centro Nazionale Ricerche, 00185 Rome, Italy; 4Department Scuola Medica Salernitana, University of Salerno, 84081 Baronissi, Italy; rachelevilla@outlook.it (R.V.); rosangelaveneziano@gmail.com (R.V.); 5Department of Epidemiology and Prevention, IRCCS Neuromed, 86077 Pozzilli, Italy; simona.costanzo@moli-sani.org (S.C.); licia.iacoviello@moli-sani.org (L.I.); 6Mediterranea Cardiocentro, 80122 Napoli, Italy; sara.magnacca@moli-sani.org (S.M.); dicastel@ngi.it (A.D.C.); 7Department of Medicine and Surgery, Research Center in Epidemiology and Preventive Medicine (EPIMED), University of Insubria, 21100 Varese, Italy

**Keywords:** sodium, proteinuria, albuminuria, epidemiology

## Abstract

Sodium effects on proteinuria are debated. This observational, cross-sectional, population-based study investigated relationships to proteinuria and albuminuria of sodium intake assessed as urinary sodium/creatinine ratio (NaCR). In 482 men and 454 women aged 35–94 years from the Moli-sani study, data were collected for the following: urinary NaCR (independent variable); urinary total proteins/creatinine ratio (PCR, mg/g), urinary albumin/creatinine ratio (ACR, mg/g), and urinary non-albumin-proteins/creatinine ratio (calculated as PCR minus ACR) (dependent variables). High values were defined as PCR ≥ 150 mg/g, ACR ≥ 30 mg/g, and urinary non-albumin-proteins/creatinine ratio ≥ 120 mg/g. Urinary variables were measured in first-void morning urine. Skewed variables were log-transformed in analyses. The covariates list included sex, age, energy intake, body mass index, waist/hip ratio, estimated urinary creatinine excretion, smoking, systolic pressure, diastolic pressure, diabetes, history of cardiovascular disease, reported treatment with antihypertensive drug, inhibitor or blocker of the renin-angiotensin system, diuretic, and log-transformed data of total physical activity, leisure physical activity, alcohol intake, and urinary ratios of urea nitrogen, potassium, and phosphorus to creatinine. In multivariable linear regression, standardized beta coefficients of urinary NaCR were positive with PCR (women and men = 0.280 and 0.242, 95% confidence interval = 0.17/0.39 and 0.13/0.35, *p* < 0.001), ACR (0.310 and 0.265, 0.20/0.42 and 0.16/0.38, *p* < 0.001), and urinary non-albumin-proteins/creatinine ratio (0.247 and 0.209, 0.14/0.36 and 0.09/0.33, *p* < 0.001). In multivariable logistic regression, higher quintile of urinary NaCR associated with odds ratio of 1.81 for high PCR (1.55/2.12, *p* < 0.001), 0.51 of 1.62 for high ACR (1.35/1.95, *p* < 0.001), and of 1.84 for high urinary non-albumin proteins/creatinine ratio (1.58/2.16, *p* < 0.001). Findings were consistent in subgroups. Data indicate independent positive associations of an index of sodium intake with proteinuria and albuminuria in the population.

## 1. Introduction

Glomerular filtration rate and proteinuria or albuminuria are essential components for the diagnosis and for the staging of chronic kidney disease [1]. Sodium intake restriction is suggested in chronic kidney disease to improve the control of hypertension and of hypertension-dependent kidney dysfunction [1]. There is not agreement on the possibility that a sodium intake restriction per se might favor the prevention and/or the control of proteinuria or albuminuria [2]. Intervention studies reported favorable effects of sodium restriction but could not dissociate the effects on blood pressure from those on albuminuria or proteinuria [2,3,4]. Observational data were inconsistent about the relationship of urinary sodium with albuminuria and were missing about the relationship of urinary sodium with proteinuria [2].

The present study was designed to analyze in a sample of the general population the associations of urinary sodium, taken as index of sodium intake, with proteinuria, albuminuria, and urinary non-albumin proteins, separately. The analyses were controlled for many confounders including socio-cultural factors, anthropometry, blood pressure status, kidney function, lifestyle, and dietary and biochemical markers.

## 2. Materials and Methods

The Moli-sani study is a prospective cohort study ongoing since 2005 that enrolled 24,325 individuals from 2005 to 2010, men and women, with age 35 and over, randomly recruited from the general population of a region of central-southern Italy [5]. The study complies with the Declaration of Helsinki of 1975, as revised in 2013, and was approved by the Rome Catholic University ethical committee (P99, A.931/03-138-04, 11 February 2004). All participants provided written informed consent. The baseline visit was conducted at the Research Laboratories of the Catholic University in Campobasso (Italy) and included the following: questionnaires about education as a proxy of socioeconomic status; questionnaires about total and leisure physical activity; the European Prospective Investigation into Cancer and Nutrition (EPIC) food frequency questionnaire specifically adapted for the Italian population for assessment of usual intake of macronutrients, micronutrients, and alcohol in the past year [6]; questionnaires about risk factors, personal and family medical history; three blood pressure measurements in the non-dominant arm by an automatic device (OMRON-HEM-705CP) with participants lying down for 5 min [5]; measurements of weight, height, and waist/hip ratio; collection of untimed urine spot samples from the first void after waking; collection of morning venous blood samples after an overnight fast. Biological samples were processed for lab tests within 3 h and/or stored in liquid nitrogen as described [7]. Lab tests for the whole cohort included the measurements of serum cystatin C [8]. Target cohort for the present analysis was a sub-cohort of 1000 examinees of the Moli-sani study who were selected by a sex- and age- stratified randomization process for additional lab tests using frozen samples of serum and urine [9]. As shown in Appendix A, the stratification was designed to have 100 men and 100 women for each one of the following five age-groups: 35–44, 45–54, 55–64, 65–74, and ≥75 years. Additional lab tests in the sub-cohort were performed by automated biochemistry and included the urinary measurements of sodium, potassium, phosphorus, urea nitrogen, total proteins, albumin, and creatinine, and the serum measurements of creatinine and total 25-hydroxy-vitamin D (25(OH)D) [9]. The measurements of urinary analytes and serum creatinine were performed using Abbott Architect c8000 (Abbott, IL, USA) [9,10]. Serum creatinine was measured by enzymatic assay calibrated with IDMS-traceable standard [11]. As stated above, original lab tests for the whole cohort included the measurements of serum cystatin C [8]. Serum 25(OH)D was measured by a fully automated chemiluminescent assay (Diasorin, Saluggia, Italy) [12]. In accordance with guidelines [13], the 25(OH)D assay was calibrated using ID-LC-MS- and ID-LC-MS/MS-traceable standard NIST-SRM 972a [14]. Intra- and inter-assay variability of all measurements was <5%.

### 2.1. Calculations

Urinary sodium was assessed as urinary sodium/creatinine ratio (NaCR) that is independent of errors in completeness and timing of timed urine collections [15]. The ratio was used as the main independent variable. The use of a single untimed morning urine sample certainly implied a misclassification of the 24 h urinary sodium due to the existence of circadian rhythms in urinary sodium excretion [15,16]. However, population-based data indicated this misclassification is low at the group level because trends along quantiles of urinary NaCR were consistent whether using 24 h urine, overnight urine, morning urine, or the untimed spot urine of different days [17]. The following three dependent variables were used in analyses: the urinary ratio of total proteins to creatinine, used as an index of proteinuria (PCR) [1]; the urinary ratio of albumin to creatinine, used as an index of albuminuria (ACR) [1]; and the difference between PCR and ACR, taken as an index of proteinuria due to non-albumin proteins. As for urinary NaCR, the use of urinary data factored by creatinine excluded errors due incomplete or imprecise urine collection. The threshold for definition of high values was ≥150 mg/g for PCR and ≥30 mg/g for ACR [1]. The threshold for definition of high urinary non-albumin proteins to creatinine ratio was ≥120 mg/g, which is the difference between the thresholds for the definition of high PCR and high ACR.

In addition to sex and age, the study included as possible covariates the following variables previously associated with kidney dysfunction: education [18], total and leisure physical activity [19], body mass index and waist/hip ratio [20], creatinine excretion [21], smoking [22], blood pressure [23], serum total cholesterol [23], serum glucose [24], cardiovascular disease history [25], estimated glomerular filtration rate (eGFR) and drug treatment [1], calorie intake [26], urinary urea nitrogen/creatinine ratio as an index of dietary protein intake [27], urinary potassium/creatinine ratio as an index of urinary potassium [28], urinary phosphorus/creatinine ratio as an index of phosphorus intake [29], alcohol intake [30], and serum 25-OH vitamin as an index of vitamin D status [31]. High education was defined as the report of high school diploma or higher. Total and leisure physical activity were expressed as metabolic equivalent of tasks per day (MET/d) [32,33]. Creatinine excretion was estimated using the CKD Epidemiology Collaboration equation [34]. Kidney function was assessed as estimated glomerular filtration rate (eGFR) calculated by the Chronic Kidney Disease—Epidemiology Collaboration equation that included serum creatinine, ethnicity, sex, age, and also cystatin C to reduce the confounding of creatinine generation [35,36]. Energy intake as kcal/d and alcohol intake as g/d were derived from the food composition database for epidemiological studies in Italy [6]. The following definitions were used for clinical characteristics: hypertension = regular drug treatment for hypertension and/or systolic pressure ≥ 140 mmHg and/or diastolic pressure ≥ 90 mmHg (means of second and third measurements); hypercholesterolemia = regular drug treatment for hypercholesterolemia and/or serum total cholesterol ≥ 240 mg/100 mL; diabetes = regular treatment with insulin or antidiabetic drugs and/or serum glucose ≥ 126 mg/dL; history of cardiovascular disease = reported previous diagnosis of myocardial infarction or stroke.

### 2.2. Statistics

For descriptive statistics, numerical variables were reported as mean ± SD and also as median and interquartile range for skewed variables (skewness < −1 or >1). Skewed variables were log-transformed in main analyses. For this transformation, non-transformed data equal to zero were also coded as zero in log-transformed data. Comparisons between sexes were done by ANOVA or chi-squared analysis.

The relationships of urinary NaCR to indices of proteinuria were first analyzed by simple linear regression using log-transformed data of skewed variables. Analyses were initially performed separately in men and women to exclude possible effect modifications due to gender-dependent differences in anthropometry, serum creatinine, urinary creatinine, and derived variables. To allow direct comparisons of the strength of the relationships, results were reported as standardized coefficient beta (beta) with 95% confidence interval (95% CI). The independence of results was assessed using multivariable linear regression with control for the following: age; physical activity; body mass index; waist/hip ratio; estimated urinary creatinine; smoking; blood pressure and antihypertensive drugs; diabetes, history of cardiovascular disease; eGFR; energy intake; the urinary ratios of urea nitrogen/creatinine, potassium/creatinine, and phosphorus/creatinine; alcohol intake; and serum vitamin D. Education, serum total cholesterol, and drug treatment for hypercholesterolemia were not included in multivariable regression because they should not have direct effects on urinary proteins or urinary creatinine (Directed Acyclic Graph in Appendix A). Categorical covariates were included in analyses as 0/1 dummy variables. Gender was added to the list of covariates in multivariable models for men and women combined.

After that, in men and women combined and with control for covariates, ANOVA was performed along quintiles of urinary NaCR to investigate the shape of the relationships with indices of proteinuria as continuous non-transformed data and with the prevalence of high urinary proteins coded as a 0/1 dummy variable. To test the possibility that findings were accounted for by an effect of urine creatinine, ANOVA was also performed for non-transformed urine concentrations as mg/L of total proteins, albumin, and non-albumin proteins by quintile of non-transformed urine sodium concentration as mmol/L (Appendix A).

Multivariable logistic regression with control for covariates was used to quantify the difference in the prevalence of high urinary proteins. Last, for analyses of the consistency of results, multivariable logistic regression was re-run in the following subgroups: men and women, age ≥ 65 years and <65 years, eGFR < 90 mL/min × 1.73 m^2^ and ≥90 mL/min × 1.73 m^2^, obese and non-obese, drinker and non-drinker, smoker and non-smoker, hypertensive and non-hypertensive, with hypercholesterolemia and without hypercholesterolemia, diabetic and non-diabetic, with cardiovascular disease and without cardiovascular disease. Statistical procedures were performed using IBM-SPSS 19.

## 3. Results

### 3.1. Descriptive Statistics

Table 1 reports descriptive data by gender in the 936 individuals with complete data for all variables. Men and women had significantly different data for all variables with the exception of age, body mass index, use of statin or insulin, and serum cystatin C. Urinary NaCR and all urinary ratios were higher in women than men due to women’s lower urinary creatinine.

Non-transformed data of urinary NaCR and all indices of proteinuria were positively skewed (Appendix A). The mode was in the intermediate values for urinary NaCR and in the lowest tail for PCR, ACR, and urinary non-albumin proteins to creatinine ratio. Prevalence of high values was higher in women for urinary PCR (women and men = 36.3% and 20.9%, *p* < 0.001), for urinary ACR (19.1% and 13.3%, *p* = 0.013), and for urinary ratio of non-albumin proteins to creatinine (37.4% and 21.9%, *p* < 0.001).

### 3.2. Linear Regression

In univariate analyses, log-transformed urinary NaCR positively related to log-transformed data of urinary PCR, urinary ACR, and urinary non-albumin proteins to creatinine ratio (upper section of Table 2). Beta values were highly significant in both sexes. In multivariable analyses, beta values of log-transformed urinary NaCR reduced but remained highly significant without differences between men and women, as indicated by overlapping 95% CI (lower section of Table 2).

Table 3 reports beta values of multiple regression in men and women combined for urinary NaCR and correlates significantly associated with at least one of the urinary proteins indices.

Independent associations were also found for systolic pressure and urinary potassium to creatinine ratio with all indices of urinary proteins, for age and urinary phosphorus with urinary PCR and urinary non-albumin proteins to creatinine ratio, for diabetes with urinary PCR and ACR, for eGFR with urinary ACR, and for serum 25-OH vitamin D with urinary ACR. The beta of urinary NaCR was at least 63–71% higher as compared with the beta of these additional correlates.

### 3.3. Quintile Analyses and Logistic Regression

ANOVA in men and women combined with control for all covariates indicated highly significant and consistent linear trends along urinary NaCR quintiles, both for absolute indices of proteinuria and for the prevalence of high values (Figure 1). Findings were similar when ANOVA was performed for non-transformed urine concentrations as mg/L of total proteins, albumin, and non-albumin proteins by quintile of non-transformed urine sodium concentration as mmol/L (Appendix A).

In multivariable logistic regression, the difference between two consecutive quintiles of urinary NaCR was associated with 1.81 odds ratio of high urinary PCR (95% CI = 1.55/2.12, *p* < 0.001), with 1.62 odds ratio of high urinary ACR (1.35/1.95, *p* < 0.001), and with 1.84 odds ratio of high urinary non-albumin proteins to creatinine ratio (1.58/2.16, *p* < 0.001). Figure 2 shows that the findings were consistent and significant in subgroup analyses for men and women, age ≥ 65 years and <65 years, eGFR < 90 mL/min × 1.73 m^2^ and ≥90 mL/min × 1.73 m^2^, obese and non-obese, drinker and non-drinker, smoker and non-smoker, hypertensive and non-hypertensive, with hypercholesterolemia and without hypercholesterolemia, non-diabetic, and without cardiovascular disease. Findings were not significant or odds ratio was not calculable in analyses on small subgroups (diabetics and individuals with cardiovascular disease).

## 4. Discussion

The study reports the original population-based findings of a cross-sectional continuous relationship of urinary sodium to total proteinuria, albuminuria, and non-albumin proteinuria independent of many variables and, in particular, of blood pressure and anti-hypertensive drug treatment. The relationship appeared significant from a clinical viewpoint and was consistent in subgroup analyses.

Main study limitations were the use of a single, untimed, morning urine sample, the lack of data for different ethnic groups, and the observational cross-sectional design. Merits of the study were the measurements both of PCR and of ACR, the use of an accurate eGFR, the calibration of the 25(OH)-vitamin D assay with international standard, and the inclusion in the analysis of many possible covariates.

Regarding previous cross-sectional population-based reports on albuminuria, the present results agree with the conclusions of all studies where sodium intake was estimated using urinary sodium [2,17]. Conversely, the results are in contrast to the conclusions of the study of Sharma et al., where sodium intake was assessed by dietary recalls [37], a method that is notoriously affected by various biases [15]. Comparisons with previous studies on PCR or urinary non-albumin proteins could not be made because, to our knowledge, this was the first analysis to consider these objectives.

Although dietary sodium intake is the main determinant of urinary sodium excretion, the use of an untimed morning urine sample instead of 24 h urine certainly implied some misclassification of individual sodium intake due to the confounding of circadian rhythms in urinary sodium excretion [15]. However, this confounding could be smaller in analyses along urinary NaCR quintiles because trends along quantiles of urinary NaCR were highly consistent between various types of partial urine collections and 24 h urine collections in the general population [17]. Given that a cross-sectional association does not necessarily imply causation, a relationship between urinary sodium and proteinuria could reflect various mechanisms: an effect of sodium intake on proteinuria, an effect of proteinuria on urinary sodium, or the existence of a “third factor” modulating urinary sodium and proteinuria in parallel. No data support an effect of proteinuria on urinary sodium or the existence of a factor modulating urine sodium and proteinuria in parallel. Therefore, at present, a reasonable mechanism appeared to be an effect of sodium intake on proteinuria, an interpretation that is supported by previous studies reporting a reduction in albuminuria after sodium intake restriction [2,3,4]. Moreover, the present results support the idea that the sodium intake effects on proteinuria were not mediated by blood pressure because the associations of urinary sodium with indices of proteinuria were all consistently independent of blood pressure and antihypertensive drugs in the whole study cohort as well as in the separate subgroups of hypertensives and non-hypertensives. An intriguing possibility is that the association of urinary sodium with indices of proteinuria could be mediated by vasopressin [38], a hormone that is secreted in response to high sodium intake [39,40] and independently associates with high albuminuria in observational and interventional studies [41,42]. The similarity of findings for albuminuria and for urinary non-albumin proteins could be explained by effects both on the glomerular permeability and/or on the protein reabsorption at the tubular level. Other mechanisms could not be excluded. From a quantitative viewpoint, the role of urinary sodium in the prevalence of proteinuria in the population was highlighted by two findings. First, the standardized beta coefficients for PCR and ACR were much higher for urinary sodium as compared with other variables. Second, the prevalence of high values of indices of proteinuria was approximately 5 times lower in quintile 1 as compared with quintile 5 of urinary NaCR.

Regarding other variables, the association with PCR and ACR of systolic pressure together with the lack of association of diastolic pressure confirmed that, for blood pressure status, systolic pressure is the stronger correlate of proteinuria or albuminuria [23]. The regression coefficient of diabetes to ACR was 2.4 times higher in comparison to PCR and 5.1 times higher in comparison with non-albumin proteins, in accordance with the concept that high albuminuria is a key marker of diabetic kidney dysfunction [43]. Moreover, the difference in the regression coefficients of diabetes with PCR and ACR indicated that the relation between PCR and ACR could differ between diabetics and nondiabetics [44]. Last, the inverse relation of serum 25(OH)-vitamin D with urinary ACR could reflect mechanism(s) similar to those proposed for anti-proteinuric effects of vitamin D supplementation [31]. Practical implications of the study concern the treatment and the prevention of proteinuria and albuminuria. For treatment, study results indicate that a dietary sodium restriction could be effective against proteinuria in various diseases because the relationships of urinary NaCR to indices of proteinuria appeared independent of blood pressure, clinical characteristics, and pharmacological treatments. For prevention, study results indicate that health-oriented policies reducing sodium intake in the population could lower the burden not only of hypertension but also of proteinuria and of related disorders.

## 5. Conclusions

This cross-sectional study reports the finding in a sample of the general population of continuous relationships of urinary sodium to total proteinuria and albuminuria independent of blood pressure status and many other variables. In addition, study results also indicate the original finding of a continuous, independent relationship of urinary sodium to urinary non-albumin proteins. Altogether, the findings also suggest that dietary sodium could have an independent role in the development of high urinary levels of total proteins, albumin, and non-albumin proteins. Therefore, dietary sodium restriction could be relevant not only for treatment but also for prevention of disorders characterized by proteinuria and or albuminuria.

## Figures and Tables

**Figure 1 nutrients-13-01255-f001:**
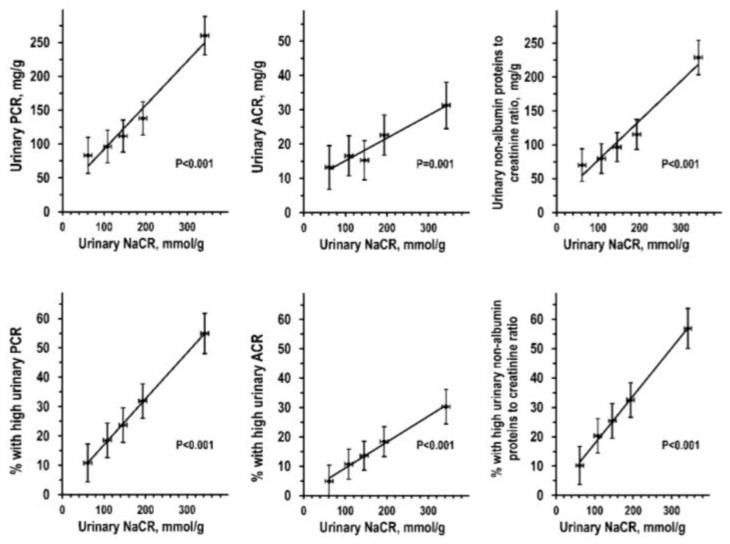
Multivariable ANOVA by urinary NaCR quintile of non-transformed data of urinary total proteins/creatinine ratio (PCR) (top left panel), urinary albumin/creatinine ratio (ACR) (top central panel), urinary non-albumin proteins to creatine ratio (top right panel), and of prevalence of high urinary PCR (bottom left panel), high urinary ACR (bottom central panel), high urinary non-albumin proteins to creatine ratio (bottom right panel): mean with 95% CI. Number of individuals from quintile 1 to quintile 5 = 191, 190, 188, 188, and 179. ANOVAs were controlled for the following covariates: age, body mass index, waist/hip ratio, estimated urinary creatinine excretion, smoking, systolic pressure, diastolic pressure, diabetes, history of cardiovascular disease, eGFR, calorie intake, reported treatment with antihypertensive drug, inhibitor or blocker of the renin-angiotensin system, diuretic, and log-transformed data of total physical activity, leisure physical activity, alcohol intake, urinary ratios of urea nitrogen to creatinine, potassium to creatinine, phosphorus to creatinine, and serum total 25(OH) vitamin D.

**Figure 2 nutrients-13-01255-f002:**
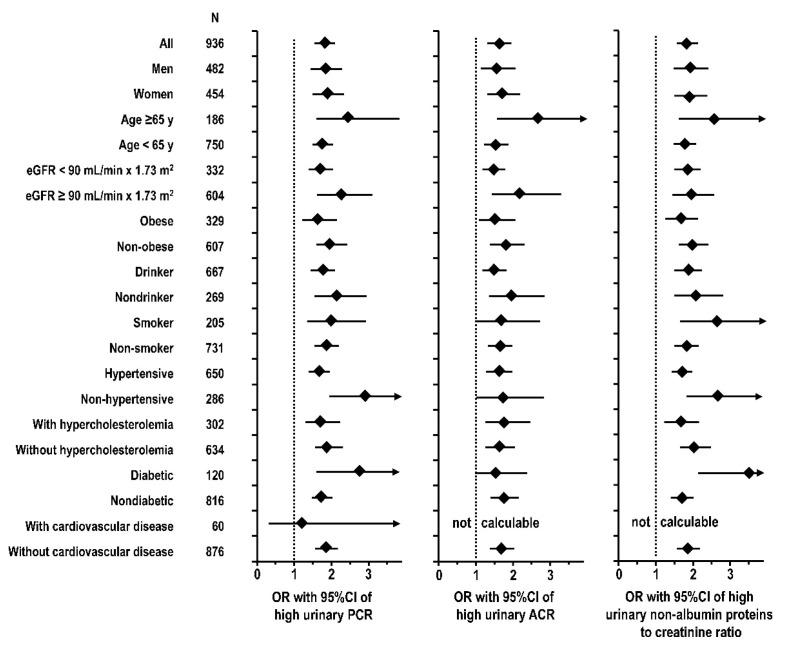
Multivariable logistic analyses of high urinary PCR, of high urinary ACR, and of high urinary non-albumin proteins to creatinine ratio alternatively regressed over urinary NaCR quintiles: odds ratio (OR) with 95% CI in the whole study cohort and in selected subgroups. Analyses were controlled for the following covariates: age, body mass index, waist/hip ratio, estimated urinary creatinine excretion, smoking, systolic pressure, diastolic pressure, diabetes, history of cardiovascular disease, eGFR, calorie intake, reported treatment with antihypertensive drug, inhibitor or blocker of the renin-angiotensin system, diuretic, statin, and log-transformed data of total physical activity, leisure physical activity, alcohol intake, urinary ratios of urea nitrogen to creatinine, potassium to creatinine, phosphorus to creatinine, and serum total 25(OH) vitamin D. Same data are presented in tabular form in Appendix A.

**Table 1 nutrients-13-01255-t001:** Descriptive statistics: mean ± SD for non-skewed variables, median for skewed variables (interquartile range), and prevalence for categorical variables.

	Women	Men	*p* ^a^
*n*	454	482	
Age, years	60.0 ± 10.0	60 ± 10	0.801
Education, % with high school or higher	38.9%	46.4%	0.021
Physical activity, MET-d			
total	40.6 (39.5–42.3)	40.7 (39.2–44.0)	
log-transformed	1.62 ± 0.05	1.63 ± 0.08	0.001
leisure	1.7 (0.3–3.4)	3.4 (1.1–6.8)	
log-transformed	0.25 ± 0.40	0.46 ± 0.46	0.001
Body mass index, kg-m^2^	28.7 ± 5.5	28.6 ± 4.2	0.793
Waist-hip ratio	0.90 ± 0.08	0.95 ± 0.06	<0.001
Estimated urinary creatinine, g-d	1.00 ± 0.17	1.52 ± 0.18	<0.001
Current smoking, %	16.1%	26.6%	<0.001
Systolic pressure, mmHg	144 ± 21	147 ± 19	0.018
Diastolic pressure, mmHg	82 ± 10	85 ± 9	<0.001
Antihypertensive drug, %	41.4%	31.9%	0.003
Inhibitor-blocker renin-angiotensin system, %	31.8%	24.6%	0.014
Diuretic, %	20.3%	12.8%	0.002
Serum total cholesterol, mg-100 mL	218 ± 39	208 ± 40	<0.001
Statin, %	9.6%	10.6%	0.697
Serum glucose, mg-100 mL	98 ± 23	107 ± 30	<0.001
Oral antidiabetic drug, %	5.4%	9.2%	0.034
Insulin treatment, %	1.5%	1.4%	0.929
Diabetes, %	9.2%	16.0%	0.002
Cardiovascular disease history, %	3.3%	9.4%	<0.001
Serum creatinine, mg-100 mL	0.74 ± 0.14	0.91 ± 0.19	<0.001
Serum cystatin C, mg-L	1.01 ± 0.23	1.03 ± 0.21	0.122
eGFR, mL-min × 1.73 m^2^	82 ± 16	85 ± 16	0.003
Energy intake, kcal-d	1844 ± 558	2227 ± 692	<0.001
Urinary creatinine, mg-100 mL	31 (16–64)	66 (31–110)	
log-transformed	1.48 ± 0.42	1.74 ± 0.43	<0.001
Urinary NaCR, mmol-g	169 (115–232)	124 (89–181)	
log-transformed	2.21 ± 0.26	2.10 ± 0.27	<0.001
Urinary urea nitrogen-creatinine ratio, g-g	9.8 (8.1–11.6)	7.7 (6.3–9.2)	
log-transformed	0.99 ± 0.13	0.88 ± 0.14	<0.001
Urinary potassium-creatinine ratio, mmol-g	82 (61–106)	63 (47–83)	
log-transformed	1.91 ± 0.17	1.80 ± 0.17	<0.001
Urinary phosphorus-creatinine ratio, mg-g	493 (364–621)	417 (296–564)	
log-transformed	2.67 ± 0.20	2.60 ± 0.216	<0.001
Alcohol intake, g-d	2 (0–12)	27 (7–48)	
log-transformed	0.48 ± 0.55	1.18 ± 0.63	<0.001
Serum 25-OH vitamin D, ng-mL	21 ± 13	22 ± 12	
log-transformed	1.25 ± 0.27	1.28 ± 0.24	0.030
Urinary total proteins-creatinine ratio, mg-g	103 (48–198)	59 (29–131)	
log-transformed	1.99 ± 0.52	1.79 ± 0.48	<0.001
Urinary albumin-creatinine ratio, mg-g	11 (5–23)	6 (3–17)	
log-transformed	1.05 ± 0.49	0.86 ± 0.56	<0.001
Urinary non-albumin proteins-creatinine ratio, mg-g	83 (37–170)	51 (23–108)	
log-transformed	1.87 ± 0.61	1.68 ± 0.53	<0.001

^a^ Comparisons between men and women by ANOVA or chi-squared analysis.

**Table 2 nutrients-13-01255-t002:** Standardized regression coefficients of log-transformed urinary sodium/creatinine ratio (NaCR) to log-transformed indices of proteinuria in simple and multivariable * regression by gender: beta with 95% CI and *p*-value.

Dependent Variable		Women *n* = 454	Men *n* = 482
Simple regression	Urinary PCR, log mg/g	0.432 (0.35/0.51) <0.001	0.369 (0.29/0.45) <0.001
	Urinary ACR, log mg/g	0.753 (0.59/0.91) <0.001	0.328 (0.24/0.41) <0.001
	Urinary non/albumin proteins to creatinine ratio, log mg/g	0.392 (0.31/0.47) <0.001	0.344 (0.26/0.43) <0.001
Multivariable regression *	Urinary PCR, log mg/g	0.280 (0.17/0.39) <0.001	0.242 (0.13/0.35) <0.001
	Urinary ACR, log mg/g	0.310 (0.20/0.42) <0.001	0.265 (0.16/0.38) <0.001
	Urinary non/albumin proteins to creatinine ratio, log mg/g	0.247 (0.14/0.36) <0.001	0.209 (0.09/0.33) <0.001

* Analyses were controlled for the following covariates: age, body mass index, waist/hip ratio, estimated urinary creatinine excretion, smoking, systolic pressure, diastolic pressure, diabetes, history of cardiovascular disease, estimated glomerular filtration rate (eGFR), calorie intake, reported treatment with antihypertensive drug, inhibitor or blocker of the renin-angiotensin system, diuretic, and log-transformed data of total physical activity, leisure physical activity, alcohol intake, urinary ratios of urea nitrogen to creatinine, potassium to creatinine, phosphorus to creatinine, and serum total 25(OH) vitamin D.

**Table 3 nutrients-13-01255-t003:** Standardized regression coefficients of variables significantly associated with indices of proteinuria in multivariable regression in men and women combined (*n* = 936): beta with 95% CI and *p*-value.

Independent Variables	Dependent Variable		
	Urinary PCR log mg/g	Urinary ACR log mg/g	Urinary Non−Albumin Proteins to Creatinine Ratio log mg/g
Age, years	−0.157 (−0.27/−0.05) 0.007	−0.121 (−0.23/−0.01) 0.035	−0.142 (−0.26/−0.02) 0.017
Systolic pressure, mmHg	0.129 (0.04/0.22) 0.005	0.170 (0.08/0.26) <0.001	0.111 (0.02/0.20) 0.018
Diabetes, yes/no = 1/0	0.064 (0.01/0.12) 0.048	0.158 (0.10/0.22) <0.001	0.026 (−0.02/0.09) n.s.
eGFR, mL/min × 1.73 m^2^	−0.077 (−0.15/0.01) n.s.	−0.094 (−0.17/−0.02) 0.022	−0.067 (−0.15/0.02) n.s.
uNaCR, log mmol/g	0.267 (0.19/0.35) <0.001	0.290 (0.21/0.37) <0.001	0.234 (0.16/0.32) <0.001
Urinary potassium/creatinine ratio, log mmol/g	0.150 (0.08/0.22) <0.001	0.116 (0.04/0.16) 0.002	0.144 (0.07/0.21) <0.001
Urinary phosphorus/creatinine ratio, log mg/g	0.075 (0.01/0.14) 0.034	0.055 (−0.02/0.12) n.s.	0.074 (0.01/0.15) 0.035
Serum 25−OH vitamin D, log ng/mL	−0.058 (−0.11/0.01) not significant	−0.075 (−0.14/−0.02) 0.017	−0.055 (−0.11/0.01) not significant

Covariates included in the model and not associated with dependent variable: gender, body mass index, waist/hip ratio, estimated urinary creatinine excretion, smoking, systolic pressure, diastolic pressure, diabetes, history of cardiovascular disease, energy intake, reported treatment with antihypertensive drug, inhibitor or blocker of the renin-angiotensin system, diuretic, and log-transformed data of total physical activity, leisure physical activity, alcohol intake, urinary ratios of urea nitrogen to creatinine, potassium to creatinine, and phosphorus to creatinine.

## Data Availability

The Moli-sani Study shared data with other studies. Data are not open access, but there are no strict limitations to data sharing. The steering committee of the study evaluates the proposals of data sharing on a case-by-case basis. Please contact licia.iacovilello@moli-sani.org for information and/or proposals.

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
