# Peer review of "Sodium Intake and Proteinuria/Albuminuria in the Population—Observational, Cross-Sectional Study"

_nutrients, 2021, doi:10.3390/nu13041255_

Round 1
Reviewer 1 Report
The authors of these analyses have studied the relations between urinary sodium excretion based on a sport urine and urinary protein excretion in a large population of subjects with or without hypertension, diabetes and other cardiovascular and renal comorbidities. Their analyses tend to confirm the associations between sodium excretion and protein excretion in this population and this even after multiple statistical corrections.
Comments:
This is an interesting analysis. However there might be some issues when assessing the correlations between urinary sodium excretion and the various proteins excretion. Indeed, all urinary parameters are corrected for urinary creatinine concentration. So the creatinine concentration in g is present both on the Y and on the X axes leading to an excellent correlation. This is the main problem of all these analyses. Did the authors try to asses the correlations using only the Na and the protein excretions without correcting for creatinine as the two measurements come from the same samples ? Do the results differ in that case ?
Reviewer 2 Report
This study examines an interesting clinical question – that of whether dietary sodium intake is associated with progression of chronic kidney disease, independent of blood pressure. However there are two main design limitations. Firstly a cross sectional study does not enable examination of time between exposure and outcome, and so a causal association cannot be demonstrated. Secondly, a single spot urine is totally inadequate as a measure of usual dietary sodium intake in individuals (1). Since the Moli-sani study is a prospective cohort study, I suggest the authors examine this question with baseline multiple 24 hour urine collections, and follow up using cohort analysis.
- Campbell NR, He FJ, Tan M, Cappuccio FP, Neal B, Woodward M, et al. The International Consortium for Quality Research on Dietary Sodium/Salt (TRUE) position statement on the use of 24‐hour, spot, and short duration (< 24 hours) timed urine collections to assess dietary sodium intake. The Journal of Clinical Hypertension. 2019.
Reviewer 3 Report
This well-written manuscript adds new information about the relationships between sodium intake and the occurrence of proteinuria and albuminuria. The authors reported that urinary sodium-to-creatinine ratio (NaCR) was associated with higher risks of proteinuria and albuminuria. However, several key aspects of the study design and data analysis remain unclear.
- Study population. It is unclear how the study participants were selected from 24,235 individuals. Also, the description of study population does not touch upon the remaining sample (N=936) for the final analyses. It would be helpful to have a flow chart showing the selection of the study population.
- Statistics. This reviewer has several concerns regarding the statistical methods.
1) Stratified analyses to exclude the possibility “confounding”. Stratification is usually used to examine effect modification. Effect modification is distinct from confounding; it occurs when the magnitude of the effect of the primary exposure on an outcome differs depending on the level of a third variable.
2) “Multivariate linear regression”. The terms “multivariate” and “multivariable” were used interchangeably throughout the manuscript. However, multivariate regression refers to the analysis of multiple outcomes whereas multivariable analysis deals with only one outcome each time.
3) “Non-transformed urinary data”. Given the highly skewed distribution of urinary NaCR, it does not make sense to use non-transformed urinary data. The authors can still get a “direct quantitative assessment of the slopes of the relationships” through log transformed NaCR (e.g., for an interquartile range change).
4) “Arbitrary difference of 100 mmol/g”. It is unclear why the authors decided to report the difference in the prevalence of proteinuria with a 100 mmol/g change in NaCR. Please be consistent and compare higher quintiles to the lowest one.
5) Please draw a DAG to justify the choice of confounders to exclude the possibility of overadjustment bias and model overfitting.
5) Please describe the sensitivity analyses in detail.
Minor comments.
- Abstract. “and many other variables”. Write out all covariates considered in the analyses.
- Introduction. First paragraph. “A sodium intake restriction per se might favor”.
- Figure 1. Put that in the supplemental materials.
Round 2
Reviewer 1 Report
I thank the authors for the additional analyses which confirm the initial ones.
I understand that authors do not want to include the new figure with the correlations not corrected for creatinine in the main paper, but it should be included as a supplemental figure.
Author Response
The authors are grateful to Reviewer 1 for his/her suggestion.
The new Figure was included in the Supplementary Material as per reviewer's request (Figure S4). Accordingly, the text of the second revision was slightly modified in METHODS (lines 165-169) and in RESULTS (lines 241-244)
Reviewer 2 Report
N/A
I stand by my previous comments that a single spot urine is inadequate for assessment of sodium intake.
Author Response
The authors are grateful to Reviewer 2 for his/her comment.
Reviewer 3 Report
The authors addressed my comments.
Author Response
The authors are grateful to Reviewer 3 for his/her comment.